# Heterogeneous organophotocatalytic HBr oxidation coupled with oxygen reduction for boosting bromination of arenes

Jie Wang[1,2], Jiahao Liang[1,2], Hao Hou[1,2], Wei Liu[1,2], Hongru Wu[1,2], Hongli Sun[1,2], Wei Ou[1,2], Chenliang Su [1,2] ✉ & Bin Liu [3,4] ✉

Developing mild photocatalytic bromination strategies using sustainable bromo source has been attracting intense interests, but there is still much room for improvement. Full utilization of redox centers of photocatalysts for efficient generation of $Br^+$ species is the key. Herein we report heterogenous organophotocatalytic HBr oxidation coupled with oxygen reduction to furnish $Br_2$ and $H_2O_2$ for effective bromination of arenes over $Al_2O_3$ supported perylene-3,4,9,10-tetracarboxylic dianhydride (PTCDA). Mechanism studies suggest that O-vacancy in $Al_2O_3$ can provide Lewis-acid-type anchoring sites for $O_2$, enabling unexpected dual-electron transfer from anchored photoexcited PTCDA to chemically bound $O_2$ to produce $H_2O_2$. The in-situ generated $H_2O_2$ and $Br_2$ over redox centers work together to generate HBrO for bromination of arenes. This work provides new insights that heterogenization of organophotocatalysts can not only help to improve their stability and recyclability, but also endow them with the ability to trigger unusual reaction mode via cooperative catalysis with supports.

Aryl bromides not only widely exist as pharmaceutical molecules, but also are essential building blocks for cross-coupling reactions in construction of valuable natural products and artificial molecules/materials[1–8]. Traditional strategies to make aryl bromides heavily relied on the use of $Br_2$ and N-bromosuccinimide (NBS), which suffered from environmental-unfriendliness, poor functionalities-tolerance, over-bromination and low atom-efficiency[2,3,5,6]. Therefore, developing sustainable and effective bromination strategies is highly imperative. Recently, homogeneous organophotocatalytic bromination of (hetero)arenes using HBr or NaBr as the bromination reagent has been emerged as a promising strategy[9,10]. However, the low-reactivity, poor photostability and difficulty of reusing the organophotocatalysts significantly restrict their widespread applications. Therefore, on one

hand, full utilization of redox centers for generation of $Br^+$ species via $Br^-$ oxidation by oxidative centers and reactive oxygen species (ROS) is crucial for improving reactivity[2]. On the other hand, heterogenization of advanced bromination organophotocatalysts not only helps to improve their photostability and recyclability, but also may introduce synergistic sites for cooperative catalysis.

Perylene-3,4,9,10-tetracarboxylic dianhydride (PTCDA) is a promising organic photoredox catalyst[11,12] with proper oxidative potential (1.73 V vs. NHE, pH = 0) and reductive potential (−0.27 V vs. NHE, pH = 0) that may realize simultaneous oxidation of $Br^-$ to $Br^+$ and reduction of $O_2$ to reactive oxygen species (ROS) for oxidative bromination[13]. Additionally, the conjugated perylene core and anhydride units of PTCDA shall enable its facile immobilization on support. As a result, in this work,

[1]International Collaborative Laboratory of 2D Materials for Optoelectronic Science & Technology, Engineering Technology Research Center for 2D Materials Information Functional Devices and Systems of Guangdong Province, Institute of Microscale Optoelectronics, Shenzhen University, Shenzhen 518060, China. [2]State Key Laboratory of Radio Frequency Heterogeneous Integration, Shenzhen University, Shenzhen 518060, China. [3]Department of Materials Science and Engineering, City University of Hong Kong, Tat Chee Avenue, Kowloon, Hong Kong SAR 999007, China. [4]Department of Chemistry, Hong Kong Institute of Clean Energy (HKICE) & Center of Super-Diamond and Advanced Films (COSDAF), City University of Hong Kong, Tat Chee Avenue, Kowloon, Hong Kong SAR 999007, China. ✉e-mail: chmsuc@szu.edu.cn; bliu48@cityu.edu.hk

PTCDA was chosen as a potential oxidative bromination organophotocatalyst. Unexpectedly, we discovered that PTCDA itself and physical mixture of PTCDA/metal oxide (i. e., $Al_2O_3$) showed little reactivity in organophotocatalytic oxidative bromination, even though superoxide radicals and $Br_2$ were generated. However, when forming strong chemical interactions between PTCDA and $Al_2O_3$ (abbreviated as PTCDA/$Al_2O_3$) verified by in-situ diffuse reflectance infrared Fourier transform spectroscopy (DRIFTs), its reactivity for photocatalytic oxidative bromination was dramatically activated. Mechanistic studies disclosed that the anhydride units of PTCDA strongly interacted with the hydroxyl groups on the surface of $Al_2O_3$ upon high-temperature treatment. The O-vacancy in $Al_2O_3$ generated during high-temperature treatment provided Lewis-acid-type anchoring sites for $O_2$. Upon light irradiation, the excited state catalyst was engaged in $Br^-$ oxidation to $Br_2$ and itself was converted to $[cat]^{\cdot-}$, which was subsequently quenched by donating an electron to $O_2$ to furnish superoxide radical. Owing to the strong interaction between $Al_2O_3$ and $O_2$, the formed superoxide radical could be fixed on the surface of $Al_2O_3$ that underwent another single electron reduction (SET) catalytic cycle to release $H_2O_2$. Finally, the in-situ generated $Br_2$ and $H_2O_2$ could work together to produce HBrO for effective bromination of arenes.

## Results and discussion

### Heterogenization of PTCDA on $Al_2O_3$ support

PTCDA has sufficient oxidative potential for $Br^-$ oxidation (1.73 V vs. NHE, pH = 0) and reductive potential (−0.27 V vs. NHE, pH = 0) for $O_2$ reduction[14–19]. To heterogenize PTCDA for providing Lewis-acid sites to adsorb organic molecules and molecular $O_2$ to enhance catalytic activity and stability, $Al_2O_3$, an insulator, was selected as the potential cooperative support. PTCDA and two-dimensional (2D) $Al_2O_3$ nanosheets were evenly ground and calcined at 320 °C to establish strong interaction (Fig. 1a). Taking 5%PTCDA/$Al_2O_3$ (5 wt.% of PTCDA) as an example, transmission electron microscope (TEM) characterizations confirm the nanosheet morphology of $Al_2O_3$ and 5%PTCDA/$Al_2O_3$ (Fig. 1b and Supplementary Fig. 1a). Meanwhile, many channels are evenly observed in the HRTEM images of 5%PTCDA/$Al_2O_3$, similar to the structure of $Al_2O_3$ as shown in Supplementary Fig. 1b, c. Furthermore, through observation with high-resolution spherical

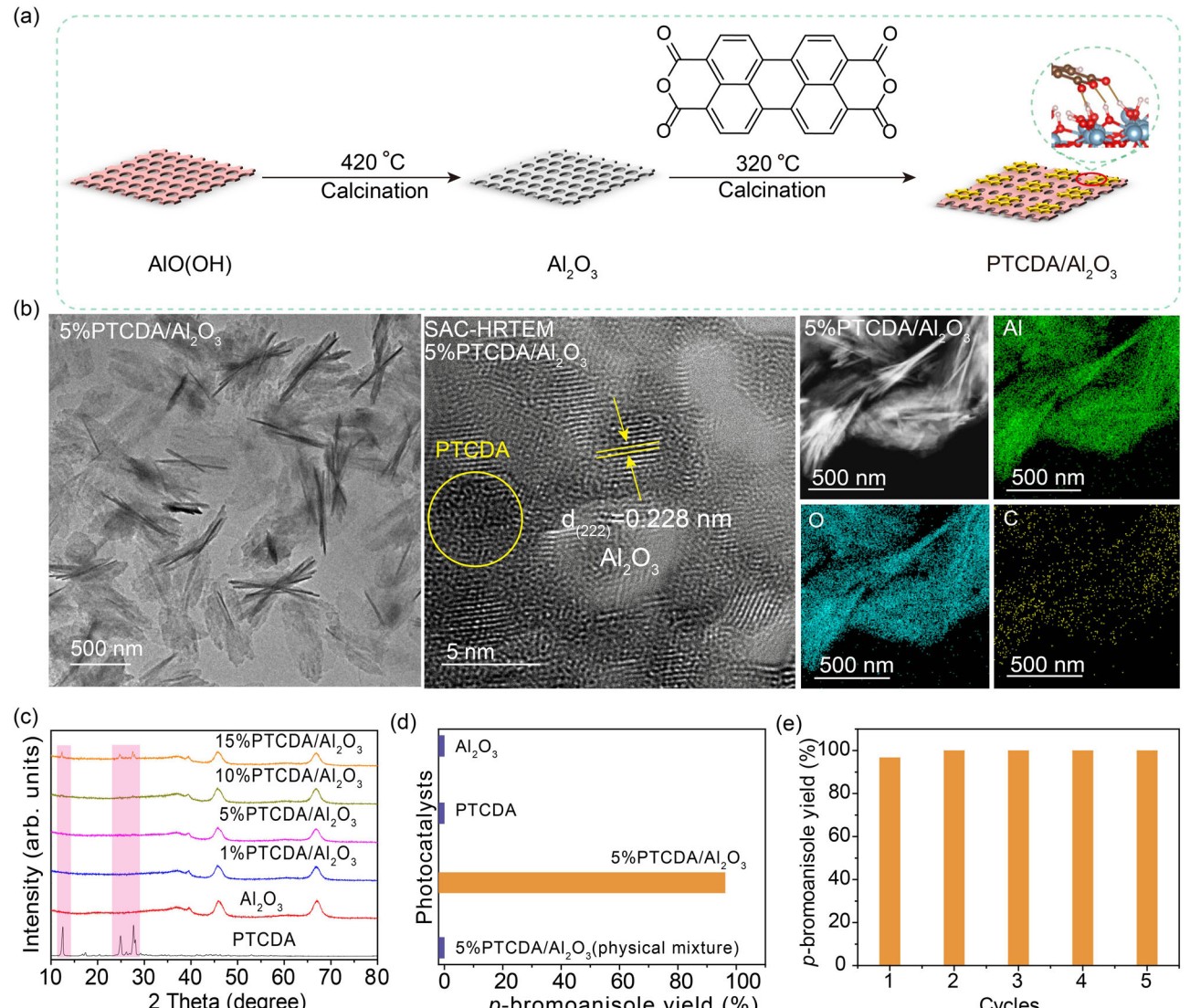

**Fig. 1 | Synthesis, characterization and photocatalytic performance.**
**a** Schematic illustration showing the preparation procedure of PTCDA/$Al_2O_3$. **b** TEM and HRTEM images, scanning transmission electron microscope (STEM) image and the corresponding elemental mappings of 5%PTCDA/$Al_2O_3$. **c** XRD patterns of PTCDA/$Al_2O_3$ with different PTCDA contents. **d** Photocatalytic bromination reaction activities of $Al_2O_3$, PTCDA, 5%PTCDA/$Al_2O_3$(physical mixture), and 5%PTCDA/$Al_2O_3$. Reaction conditions: 10 mg of photocatalyst; temperature: 30 °C; reaction time: 5 h; acetonitrile: 1 mL; anisole: 0.2 mmol; HBr: 0.2 mL; $O_2$: 1 atm; blue LED: 100 mW cm$^{-2}$. **e** Evaluation of photocatalytic stability.

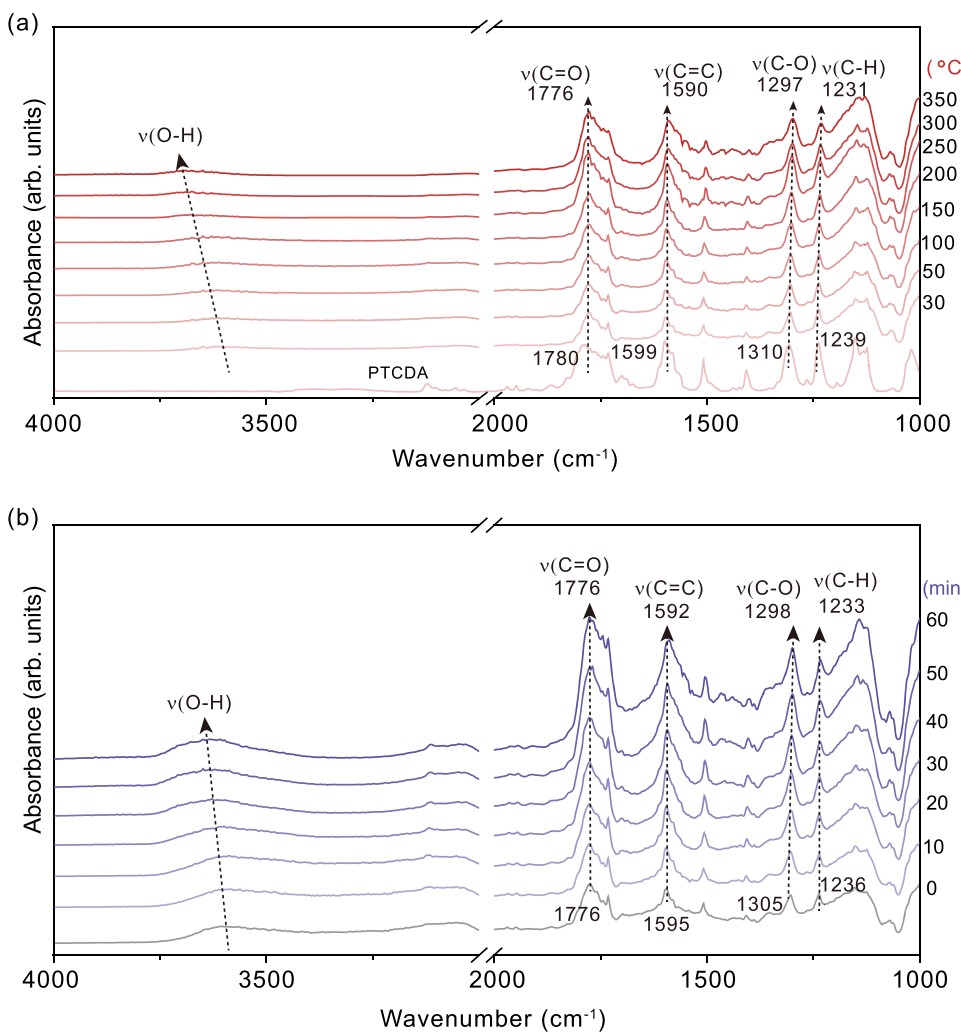

**Fig. 2 | Interaction between PTCDA and Al₂O₃. a** In-situ DRIFTs spectra of PTCDA and Al₂O₃ mixture calcined at different temperatures. **b** In-situ DRIFTs spectra of PTCDA and Al₂O₃ mixture calcined for different time durations.

aberration corrected TEM, PTCDA can be identified in 5%PTCDA/Al₂O₃, which is well-anchored on the surface of Al₂O₃. Besides, the 0.228 nm lattice spacings can be assigned to Al₂O₃ (222) plane, matching well with the X-ray diffraction (XRD) pattern (Fig. 1c)[20]. Energy dispersive X-ray spectroscopy (EDS) mappings show uniform distribution of the Al, C and O elements, demonstrating homogeneous attachment of PTCDA on Al₂O₃ surface. XRD patterns show gradual increase of PTCDA intensity with increasing PTCDA content in PTCDA/Al₂O₃ while the diffraction peaks of Al₂O₃ remain nearly constant.

To understand the support effect, Al₂O₃, PTCDA, their physical mixture, and PTCDA/Al₂O₃ were studied for bromination of arenes under visible light illumination and the results are shown in Supplementary Fig. 2 and Fig. 1d. PTCDA and PTCDA/Al₂O₃(physical mixture) show low reactivities in oxidative bromination reaction, while 5% PTCDA/Al₂O₃ displays excellent oxidative bromination performance with 95% yield (the actual mass percentage of PTCDA in 5%PTCDA/Al₂O₃ was determined to be around 5.0 wt.% by elemental analyzer as shown in Supplementary Table 1). More impressively, this organophotocatalyst shows excellent reusability and its reactivity can keep almost unchanged after multiple runs of bromination experiments (Fig. 1e and Supplementary Fig. 2).

**Interaction between PTCDA and Al₂O₃**

To explore how simple heat treatment of PTCDA and Al₂O₃ mixture could dramatically improve photocatalytic bromination reaction

performance, Fourier transform infrared (FTIR) spectroscopy and solid-state ¹³C NMR measurements were performed to investigate the structure change of PTCDA before and after calcination over Al₂O₃. As shown in Supplementary Fig. 3, the fingerprint peaks of PTCDA in both FTIR and CNMR spectra are well kept before and after thermal treatment but with slight shift, suggesting that PTCDA remained unchanged in structure after thermal treatment and formed strong interaction with Al₂O₃. Thermal gravimetry analysis (TGA) (Supplementary Fig. 4) confirms thermostability of PTCDA and PTCDA/Al₂O₃ below 400 °C. To gain more insights, in-situ diffuse reflectance infrared Fourier transform spectroscopy (DRIFTs) were conducted (Fig. 2). Fig. 2a shows that during calcination process, the vibration peaks of C=O, C=C, C–O and C–H of PTCDA apparently red shift, while the broad OH stretching vibration peak originated from Al₂O₃ significantly blue shifts; these vibration peaks are in great contrast to that of sole PTCDA (Supplementary Fig. 5a) or Al₂O₃ (Supplementary Fig. 5b), validating formation of strong interaction between PTCDA and surficial hydroxyl group on Al₂O₃[21], which is further confirmed by heating PTCDA and Al₂O₃ mixture at 320 °C for different durations (Fig. 2b and Supplementary Fig. 5c).

The interaction mode of PTCDA and Al₂O₃ was further characterized by X-ray photoelectron spectroscopy (XPS) (Fig. 3a, b). After calcining PTCDA with Al₂O₃, the O 1s binding energy of C=O and C–O in PTCDA slightly shifted from 531.5 to 531.2 eV and from 533.4 to 532.4 eV, respectively[22]. The C 1s binding energy of C=O in PTCDA

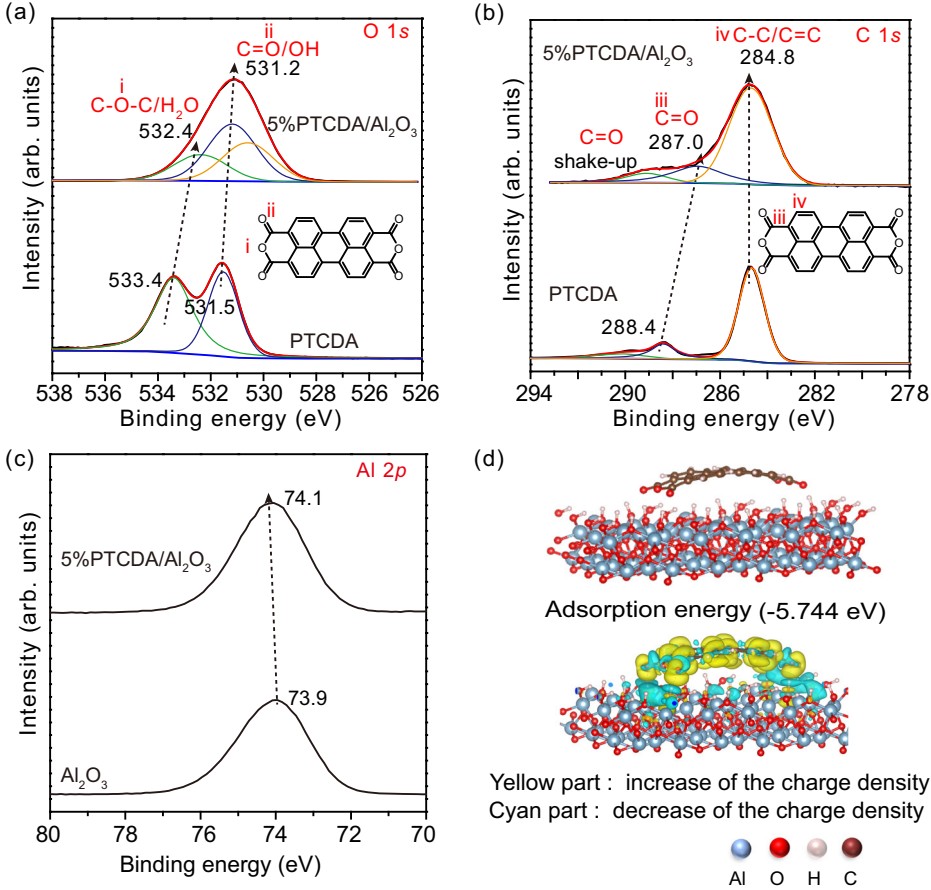

**Fig. 3 | Anchoring mode between PTCDA on Al$_2$O$_3$ surface.** High-resolution XPS spectra of (**a**) O 1$s$ and (**b**) C 1$s$ for 5%PTCDA/Al$_2$O$_3$ and PTCDA. **c** High-resolution Al 2$p$ XPS spectra for 5%PTCDA/Al$_2$O$_3$ and Al$_2$O$_3$. **d** The adsorption mode of PTCDA on Al$_2$O$_3$ surface and the differential charge of PTCDA on Al$_2$O$_3$ surface. The isovalue is $1.25 \times 10^{-3}$ eÅ$^{-3}$. The yellow/cyan region represents increase/decrease in charge density.

accordingly shifted from 288.4 to 287.0 eV[16], but one of the C 1$s$ for C–C/C=C bond had little change. Moreover, the binding energy of Al 2$p$ shifted from 73.9 to 74.1 eV (Fig. 3c)[20]. These results indicate that the O or C charge density of C–O and C=O increases while the Al charge density decreases after forming a strong interaction between PTCDA and Al$_2$O$_3$ via thermal treatment, which can be attributed to the electron withdrawing effect of anhydride units in PTCDA. It also suggests that the interaction between PTCDA and Al$_2$O$_3$ is mainly formed via O of C–O and C=O in PTCDA with surface OH on Al$_2$O$_3$. To further probe the electron state, the PTCDA adsorption configuration on Al$_2$O$_3$ was calculated and differential charge analysis is performed (Fig. 3d). After attached on Al$_2$O$_3$ surface, PTCDA molecule exhibited an increase in charge density, which is consistent with the results obtained by XPS. Meanwhile, other adsorption modes, differential charge and Bader charge as well as negative-projected crystal orbital Hamilton population (−pCOHP) for the O and H atoms with the shortest bond length were also investigated (Supplementary Fig. 6). The calculated adsorption energy is −4.808 eV for model 2 and −3.356 eV for model 3, more positive than that for model 1 (−5.744 eV), and the integral COHP values are −2.13324 for model 1, −1.85124 for model 2, and −1.44866 for model 3. These results indicate the strongest interaction between PTCDA and Al$_2$O$_3$ in model 1.

Supplementary Fig. 7a compares the UV-vis absorption spectra for PTCDA, Al$_2$O$_3$, 5%PTCDA/Al$_2$O$_3$(physical mixture), and 5%PTCDA/Al$_2$O$_3$. It can be found that the absorption spectrum of 5%PTCDA/Al$_2$O$_3$ blue shifted in the visible range, while there was almost no change for the absorption spectrum of 5%PTCDA/Al$_2$O$_3$(physical mixture) as compared to that of PTCDA. Since Al$_2$O$_3$ only absorbs UV light

(<400 nm)[23], the blueshift of light absorption for 5%PTCDA/Al$_2$O$_3$ might result from hybridization of electronic orbitals formed between PTCDA and Al$_2$O$_3$. The absorption edge of PTCDA and 5%PTCDA/Al$_2$O$_3$(physical mixture) are at around 639 nm (bandgap: 1.94 eV) and 631 nm (bandgap: 1.96 eV), respectively[17], while the absorption edge of 5%PTCDA/Al$_2$O$_3$ is at about 681 nm (bandgap: 1.82 eV). According to the published data[17,24,25], the reduction potential of PTCDA is at −0.27 V vs. NHE (pH = 0) and the conduction band edge of Al$_2$O$_3$ is at −0.30 V vs. NHE (pH = 0), which are more negative than the potential of O$_2$/O$_2$·⁻ [−0.13 V vs. NHE (pH = 0)] and O$_2$/O$_2$²⁻ [0.68 V vs. NHE (pH = 0)], implying that 5%PTCDA/Al$_2$O$_3$(physical mixture) and 5%PTCDA/Al$_2$O$_3$ are able to realize O$_2$ activation and thus produce ROS under light illumination (Supplementary Fig. 7b). To further understand the O$_2$ activation process, femtosecond transient absorption spectroscopy spectra were recorded under the condition of air saturation (Supplementary Fig. 7c, d)[26]. There shows an obvious broad positive signal in the wavelength range from 550 to 750 nm over 5%PTCDA/Al$_2$O$_3$ (Supplementary Fig. 7c). As time went from 0 to 1 ps, the intensity of the positive signal gradually increased to a maximum. Then, the signal intensity slightly decreased at t = 10 ps and maintained stable during 10−30 ps. Afterwards, the signal gradually reduced to 0 in the time range from 30 to 1000 ps. The increase of the positive signal results from accumulation of photogenerated electrons in the excited states. After reaching maximum, the slight signal intensity decrease (1−10 ps) and short maintenance of balanced states (10−30 ps) can be attributed to O$_2$ activation by the excited electrons. The signal in the wavelength range from 550 to 750 nm was much weaker over 5%PTCDA/Al$_2$O$_3$(physical mixture) (Supplementary Fig. 7d).

## Photocatalytic mechanism

Control experiments under different reaction atmospheres pinpoints the important role of $O_2$ for photocatalytic bromination of arenes over PTCDA/$Al_2O_3$ (Supplementary Table 2, entry 1-2). To identify the key reactive species, scavenger studies were carefully designed and carried out, butylated hydroxytoluene (BHT), nitrotetrazolium blue chloride (NBT), *t*-butyl alcohol (*t*-BuOH) and 9,10-diphenylanthrene (DPA) are used as the sacrificial agents to probe the roles of carbon radical (R•), superoxide radical (•$O_2^-$), hydroxyl radical (•OH) and singlet oxygen ($^1O_2$), respectively[27], and the results are shown in Supplementary Fig. 8a. When NBT or *t*-BuOH was added into the reaction system, the yield of *p*-bromoanisole decreased from 95% to 28% or 43%, indicating that •$O_2^-$ and •OH played critical roles in the photocatalytic bromination reaction. On the other hand, addition of BHT or DPA had little influence on the photocatalytic performance. To directly probe the ROS, electron paramagnetic resonance (EPR) experiments were conducted. Under light irradiation with continuous bubbling of $O_2$ (Supplementary Fig. 8b), intense EPR signals of •$O_2^-$ were observed over PTCDA, 5%PTCDA/$Al_2O_3$ and 5%PTCDA/$Al_2O_3$(physical mixture), but not over $Al_2O_3$[28]. In consideration of the low reactivities of PTCDA and PTCDA/$Al_2O_3$(physical mixture) as shown in Fig. 1d, based on the results of scavenger studies, it is concluded that •$O_2^-$ species is essential but not the only prerequisite for the photocatalytic bromination reaction. •$O_2^-$ might work as an intermediate to generate other ROS, for example $H_2O_2$ can be generated via SET reduction of •$O_2^-$. 0.009 mmol of $H_2O_2$ (quantified by iodometry method) was observed without adding HBr as the bromine source in the reaction system of 5% PTCDA/$Al_2O_3$ (Supplementary Table 3, entry 1), while no $H_2O_2$ could be detected in the reaction system of PTCDA and 5%PTCDA/$Al_2O_3$(physical mixture) (Supplementary Table 3, entry 2-3). To investigate the role of $H_2O_2$ in oxidative bromination reaction with HBr, 0.25 mmol of $H_2O_2$ was purposely added into the reaction system in the absence of photocatalyst and light, 96% yield of *p*-bromoanisole was obtained (Supplementary Table 2, entry 5), suggesting that $H_2O_2$ worked together with HBr in the oxidative bromination of arenes. $H_2O_2$ can be generated via $2e^-$ photocatalytic oxygen reduction reaction or $2e^-$ photocatalytic water oxidation reaction[29–35]. To verify the reaction pathway of $H_2O_2$, electron scavenger ($H_2PtCl_6$) was added into the reaction system under Ar atmosphere, and no $H_2O_2$ could be detected, which excludes the $2e^-$ photocatalytic water oxidation pathway to generate $H_2O_2$. Control experiments under $O_2$ and Ar atmosphere as shown in Supplementary Table 2 (entries 1-2) and EPR studies as shown in Supplementary Fig. 9 verify that the $H_2O_2$ was produced via oxygen reduction reaction during the photocatalytic bromination reaction.

Most organophotocatalytic reactions undergo oxidative and reductive quenching cycles via single electron transfer (SET) processes. Herein we discover that the heterogeneous PTCDA/$Al_2O_3$ can enable a different dual-electron transfer process to generate $H_2O_2$ instead of the traditional SET. To shed light on this unusual phenomenon, in-situ DRIFTs was performed to probe the oxygen reduction reaction under visible light irradiation[36,37]. As shown in Fig. 4a, adsorption of •$O_2^-$ at 1029 $cm^{-1}$ can be clearly observed over 5%PTCDA/$Al_2O_3$(physical mixture). Interestingly, the -O-O- peak at 797 $cm^{-1}$ is more intensive than the •$O_2^-$ peak over 5%PTCDA/$Al_2O_3$. These results suggest that the $O_2$ reduction behavior over 5%PTCDA/$Al_2O_3$ is different from that over PTCDA and 5%PTCDA/$Al_2O_3$(physical mixture). To experimentally probe $O_2$ adsorption, $O_2$-TPD (temperature programmed desorption) experiments were conducted. As shown in Fig. 4b, the $O_2$ adsorption capacity (chemical adsorption) over 5% PTCDA/$Al_2O_3$ is much higher than that over PTCDA and 5%PTCDA/$Al_2O_3$(physical mixture). EPR measurements suggest a much higher concentration of oxygen vacancies in 5%PTCDA/$Al_2O_3$ (Fig. 4c), which were possibly generated during the heat-treatment of $Al_2O_3$ and PTCDA (demonstrated in Supplementary Fig. 10a, b). The rich surface oxygen vacancies on 5%PTCDA/$Al_2O_3$ shall provide abundant

adsorption sites to chemically activate $O_2$ molecules, thus promoting two-electron transfer from the firmly anchored photoexcited PTCDA to chemically bound $O_2$ under light irradiation to produce $H_2O_2$. Theoretical calculations further support enhanced $O_2$ adsorption on PTCDA/$Al_2O_3$(Model 1) ($\Delta G_{*O_2} = -2.381$ eV, Fig. 4e) as compared to that on PTCDA/$Al_2O_3$(physical mixture, Model 3) ($\Delta G_{*O_2} = -1.735$ eV, Supplementary Fig. 10c).

Next the contributions of $H_2O_2$ and $Br_2$ in oxidative bromination reaction were studied. According to literature[2,5,38], $H_2O_2$ is able to oxide HBr to generate active electrophilic HBrO. $Br_2$ can react with water to form HBrO as well. The HBrO can subsequently react with the substrate in an electrophilic substitution reaction. First, the generation of $Br_2$ was well demonstrated by conducting the photocatalytic reaction in the presence of electron scavenger ($H_2PtCl_6$) under Ar atmosphere, during which HBr oxidation and generation of $Br_2$ (at 380 nm) could be clearly observed in the UV-vis absorption spectra (Supplementary Fig. 11a, b). Then, $H_2O_2$ or $Br_2$ was directly added into the reaction system in absence of light and photocatalyst to investigate their roles in oxidative bromination. As shown in Fig. 4d, with increasing amount of added $H_2O_2$ or $Br_2$, the yield of *p*-bromoanisole increases. Meanwhile, to verify the role of HBrO, HBrO was purposely added into the reaction system without photocatalyst in dark. Addition of 0.25 mmol HBrO resulted in a *p*-bromoanisole yield of 95%, confirming the contribution of HBrO in the bromination reaction (Supplementary Table 2, entry 6). With these results, it is concluded that the major roles of the photoredox centers in PTCDA/$Al_2O_3$ are to furnish $Br_2$ and $H_2O_2$, which work synergistically to generate active electrophilic HBrO for sequential bromination.

Figure 4e proposes the possible photocatalytic bromination reaction mechanism over PTCDA/$Al_2O_3$ photocatalyst. Under light irradiation, the excited state PTCDA* oxidizes $Br^-$ to produce $Br_2$ and PTCDA•[-] via a SET process[18]. The resultant PTCDA•[-] undergoes a reductive quenching cycle with $O_2$ to regenerate PTCDA, meanwhile $O_2$ obtains one electron and one proton to form •OOH, which is firmly fixed on the surface of photocatalyst. Next, the •OOH takes one more electron and one more proton to make $H_2O_2$. According to the reported literatures and our experimental data[2,5], the reaction between $Br_2$ and water or $H_2O_2$ and $Br^-$ can produce active electrophilic HBrO species, which will spontaneously react with arenes to produce brominated products.

## Substrate scopes

The photocatalytic oxidative bromination reaction developed here shows good extensibility to a large family of arenes (Fig. 5)[39–46]. Electron-rich arenes bearing one, two and three substituents are found to be well tolerated with corresponding products in good to excellent yields. Bromination of heteroarenes has also been demonstrated by using 1-phenylpyrazole as the model substrate, giving the desired product in 71% yield. Arenes with sensitive functional groups including secondary amine, tertiary amine, halides and heterocycles are compatible with the photocatalytic oxidative bromination reaction system. Taking carbazoles as substrates, their brominated products, which can be used as building blocks for making organic photoelectronic materials, could be obtained in 80–92% yields. For the purpose of practical application, a gram-scale flow system[4,47] using 1,2,3-trimethoxybenzene as the model substrate was designed as shown in Fig. 5, which could produce brominated product in 82% yield.

In summary, we have demonstrated that heterogenization of organophotocatalyst such as PTCDA can dramatically switch on the reactivity in photocatalytic oxidative bromination reaction using HBr as the atom-economic bromination reagent. This mild photocatalytic bromination strategy shows good extensibility, recyclability and scalability. Mechanism studies discover that heat treatment of PTCDA and $Al_2O_3$ can not only establish strong chemical interactions between them to facilitate electron transfer, but also introduce O vacancies on

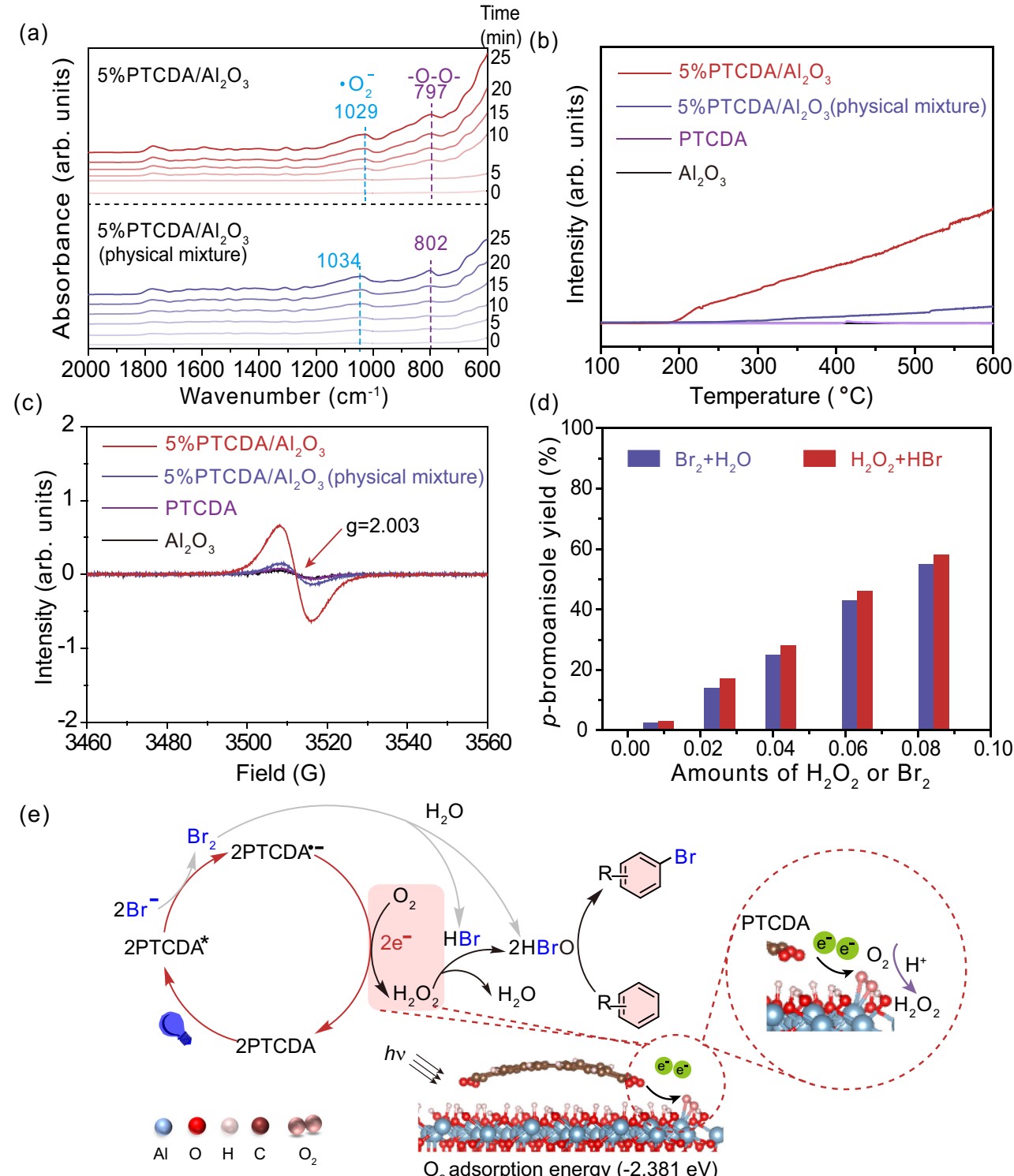

**Fig. 4 | Photocatalytic mechanism. a** In-situ DRIFTs spectra recorded over 5% PTCDA/Al$_2$O$_3$ and 5%PTCDA/Al$_2$O$_3$(physical mixture) under visible light irradiation for different time durations. **b** O$_2$-TPD and **c** EPR spectra of Al$_2$O$_3$, PTCDA, 5% PTCDA/Al$_2$O$_3$(physical mixture), and 5%PTCDA/Al$_2$O$_3$. **d** The influence of H$_2$O$_2$ or Br$_2$ on photocatalytic bromination reaction. temperature: 30 °C; reaction time: 5 h; acetonitrile: 1 mL; anisole: 0.2 mmol; Air: 1 atm; blue LED: 100 mW cm$^{-2}$. In the system of Br$_2$ + H$_2$O, HBr was not added, but in H$_2$O$_2$ system, HBr was added. The amounts of H$_2$O$_2$ or Br$_2$ added are 0.0042, 0.0084, 0.0253, 0.0422, 0.0633, and 0.0843 mmol, respectively. **e** The possible photocatalytic bromination reaction mechanism.

Al$_2$O$_3$ to adsorb molecular O$_2$. Benefiting from these synergistic effects, an unusual dual-electron transfer reaction mode is newly presented, where O$_2$ is reduced to H$_2$O$_2$ on the photoreductive centers. Meanwhile, HBr oxidation occurs on photooxidative centers to furnish Br$_2$. Both Br$_2$ and H$_2$O$_2$ contribute for the effective bromination of arenes. This work highlights that heterogenization of organophotocatalysts exhibits great potential for improving their stability and recyclability, and more importantly, it is able to trigger different reaction mode via synergistic catalysis with the well-designed supports.

**Fig. 5 | The expanding of substrate scope for photocatalytic bromination and the implementation of fluid-phase reaction using 5%PTCDA/Al₂O₃ as a photocatalyst.** The yields of brominated products with anisole, *o*-methylanisole and *m*-methylanisole as substrates are quantified by gas chromatography-mass spectrometer (GC-MS) with *p*-fluoronitrobenzene as the internal standard. The yields of other brominated products are quantified by isolated yield. Fluid-phase reaction conditions: room temperature; time: 4 h; acetonitrile: 90 mL; substrate: 1 g; HBr: 10 mL; O₂: 1 atm; blue LED:50 mW cm⁻²; and flowrate: 10 mL min⁻¹.

## Methods

### Chemicals

Aluminum nitrate nonahydrate (Al(NO₃)₃·9H₂O), urea, 3,4,9,10-perylenetetracarboxylic dianhydride (PTCDA, 98%), triethylamine hydrochloride, anisole, acetonitrile, and hydrogen bromide (HBr, 48% in H₂O) were purchased from Energy Chemical and used without further purification.

### Preparation of 2D Al₂O₃ nanosheets

2D Al₂O₃ nanosheets were prepared through a hydrothermal method[21,48,49]. In a typical synthesis, 4.5 g of Al(NO₃)₃·9H₂O and 5.1 g of urea were added to a 250 mL round bottom flask, followed by adding 200 ml of ultrapure water. Then the obtained mixture was maintained at 150 °C for 24 h. After cooling, the white product was washed to neutral using ultrapure water, and then dried in an oven at 75 °C for 24 h to obtain AlO(OH). AlO(OH) was calcined under air atmosphere for 8 h at 420 °C to make Al₂O₃ nanosheets.

### Preparation of PTCDA/Al₂O₃

For a typical synthesis of 5%PTCDA/Al₂O₃, 0.3 g of Al₂O₃ and 0.0158 g of PTCDA were ground evenly and then added into a quartz boat, which was calcined at 320 °C for 1 h inside a tube furnace under air atmosphere at a heating rate of 5 °C/min. After cooling down to room temperature, the obtained sample was washed twice with 0.1 M triethylamine hydrochloride solution, followed by three times with ultrapure water and once with ethanol, and then dried at 60 °C in a vacuum oven for 12 h to obtain a composite of PTCDA and Al₂O₃, labeled as 5% PTCDA/Al₂O₃. PTCDA/Al₂O₃ with different PTCDA mass percentages

were obtained using the same method by only changing the mass of PTCDA.

## Characterizations

Transmission electron microscopy (TEM) measurements were carried out on a JEOL F200 microscope with accelerating voltage of 200 kV. The high-resolution transmission electron microscope images were obtained on a JEOL JEM-ARF200F TEM/STEM equipped with a spherical aberration corrector. The XRD patterns were recorded on a Rigaku Ultima IV X-RAY diffractometer with Cu Kα radiation ($\lambda = 1.5418$ Å) in the $2\theta$ range from 5 to 80° at a scanning rate of 8° $min^{-1}$. X-ray photoelectron spectroscopy (XPS) and in-situ XPS under light irradiation were conducted on a Escalab 250Xi spectrometer at room temperature using an Al Kα X-ray source ($hv = 1486.6$ eV). The C $1s$ peak at 284.8 eV was used as the reference for the calibration of the binding energy. UV-vis diffuse reflectance spectra were measured on an Agilent Technologies Cary Series UV-vis-NIR spectrometer. In-situ diffuse reflectance infrared Fourier transform spectroscopy (DRIFTs) measurements were conducted on a Bruker Tensor II spectrometers. The percentage of PTCDA was determined by an organic element analyzer on elementar EL III. Thermogravimetric/differential thermal (TG/DTA) analysis was carried out on a Thermo plus EV2 in air with a heating rate of 10 °C $min^{-1}$. $^1H$ and $^{13}C$ NMR spectra were recorded by a Bruker AVANCE III spectrometer (frequencies of 600 and 100 MHz) and the chemical shift was referenced to TMS (tetramethylsilane).

## Photocatalytic bromination reaction

Typically, 10 mg of photocatalyst, 1 mL of anisole in acetonitrile solution (0.2 M) and 0.2 mL of HBr solution were added into a 10 mL vial. The vial was then purged with $O_2$ for 1 min to maintain the $O_2$ pressure at 1 atm and kept at 30 °C in an oil bath for 5 h under visible light irradiation [Blue LED (center wavelength, 460 nm), 100 mW $cm^{-2}$]. Afterwards, 0.01 mmol of $p$-fluoronitrobenzene as the internal standard and 3 mL of ultrapure water were added to the vial. The products were extracted using 1 mL of ethyl acetate. The supernatant was analyzed using Agilent Technologies 7820 gas chromatography equipped with a WondaCap 5 column. The amounts of products and reactants were calculated using an internal standard method.

## $H_2O_2$ production and detection

The $H_2O_2$ concentration in solution was determined by an iodometry method as reported[50]. After reaction, 0.1 mL of reaction solution was centrifuged, followed by adding 1 ml of potassium titanium oxalate solution (0.1 mol $L^{-1}$), 1 mL of KI solution (0.4 M) and 4 ml of deionized water. The concentration of $H_2O_2$ was determined based on the absorbance at 350 nm in the UV-vis spectra.

## EPR experiments

To capture •$O_2^-$, 10 mg of photocatalyst, 1 mL of methanol, and 50 μL of 5,5-dimethyl-1-pyrroline N-oxide (DMPO) solution (methanol: DMPO = 10: 1) were added into a photocatalytic reactor and then irradiated by visible light under $O_2$ atmosphere. 20 μL of solution was taken out for EPR measurement after different durations of photocatalytic reaction. To capture •OH, 10 mg of photocatalyst, 1 mL of $H_2O$, 0.2 mmol of $H_2PtCl_6$ and 50 μL of 5,5-dimethyl-1-pyrroline N-oxide (DMPO) solution (acetonitrile: DMPO = 10: 1) were added into a photocatalytic reactor and then irradiated by visible light under Ar atmosphere. 20 μL of solution was taken out for EPR measurement after different durations of photocatalytic reaction. Room-temperature EPR spectra were recorded on Bruker EMXPLUS10/12 EPR electron paramagnetic resonance spectrometer.

## TPD experiments

Oxygen desorption was measured by temperature programmed desorption (TPD) of $O_2$ in a micro-reactor. Typically, 100 mg of catalyst was added to a micro-reactor, pre-treated in Ar at 200 °C for 1 h and then cooled to 30 °C. The adsorption of $O_2$ (30 mL $min^{-1}$) was implemented for 10 min at 30 °C and then the catalyst was flushed with Ar (30 mL $min^{-1}$) for 30 min at 30 °C to remove the physically adsorbed gas on the surface of the catalyst. Programmed desorption was performed at a heating rate of 10 °C $min^{-1}$ from 50 to 800 °C probed by a TCD detector.

## Computational details

The spin-polarized density functional theories (DFT) were carried out by using the Vienna Ab initio Simulation Package (VASP)[51]. (110) surface was chosen as the active surface to represent the as-prepared γ-$Al_2O_3$ in our calculation model because that it is estimated that (110) surface is the predominant exposed-surface according to the Gibbs–Curie–Wulff law[52], which occupies 74% of the total surface area, followed by (100) surface (16%) and (111) surface (10%). The Perdew-Burke-Ernzerhof generalized-gradient approximation functional was used to describe the interaction between electrons[53]. The DFT-D2 method was adopted to evaluate the van der Waals (vdW) interaction[54]. The energy cutoff was set to 400 eV. The energy criterion was set to $10^{-5}$ eV in the iterative solution of the Kohn-Sham equation. The Brillouin zone integration was performed using a $2 \times 2 \times 1$ k-mesh. All the structures are relaxed until the residual forces on the atoms have declined to less than 0.02 eV/Å.

## Data availability

The data generated in this study are provided in the article and the Supplementary Information file and are available from the corresponding author upon request. Source data are kept in the repositories https://doi.org/10.6084/m9.figshare.25669848.

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

## Acknowledgements

This work was supported by The National Key Research and Development Program of China (2021YFA1600800), National Natural Science Foundation of China (21972094, 22102102, 22372102), Educational Commission of Guangdong Province (839-0000013131), Shenzhen Science and Technology Program (RCJC20200714114434086), the City University of Hong Kong startup fund (9020003), ITF-RTH – Global STEM Professorship (9446006), the Guangdong Basic and Applied Basic Research Foundation (2020A1515010982), and Research Team Cultivation Program of Shenzhen University (2023QNT013). The authors also thank Dr. Nan Jian from the Electron Microscope Center of Shenzhen University for his help with TEM measurements and SuanChou (suanchou.com) for the theoretical study.

## Author contributions

J. Wang, C. L. Su and B. Liu designed research, analyzed data and wrote the paper. J. H. Liang, H. Hou, W Liu, H. R. Wu, H. L. Sun and W. Ou analyzed data.

## Competing interests

The authors declare no competing interests.
