## [Peer Review File · Nature Communications]

Heterogeneous Organophotocatalytic HBr Oxidation Coupled with Oxygen Reduction for Bromination of ArenesREVIEWER COMMENTS

Reviewer #1 (Remarks to the Author):

The reviewed article comprises a few novelties:

- Rationalization of the photocatalytic mechanism of the bromination by HBr catalysed by the PTCDA/Al₂O₃ system.
- The heat treatment of the mixture of PTCDA and Al₂O₃ was found necessary for the formation of O vacancies and the formation of the covalent bonding between these subsystems

The Authors competently used many complementary experimental techniques. The experimental part is of high quality. The Authors suggest an unexpected mechanism (dual-electron transfer process to generate H₂O₂ rather than single electron transfer) and their reasoning is well-supported and documented.

In my opinion, the Article deserves publishing in case the below-mentioned questions are addressed.

As for the experimental part, a question may emerge about the geometry details of the adsorption mode (see below).

A few objections may be posed for the computational part, namely:

For the adsorption mode of PTCDA on the Al₂O₃ surface:

- 1) Al₂O₃ termination - how could the Authors convince the reader that the termination chosen by the Authors is the most represented?
- 2) Did the Authors try the other adsorption modes/conformation? If not, what is the reasoning of the chosen one? The HR-TEM images (Fig. 1 b) do not show any periodic arrangement.
- 3) Did the Authors further study the interaction between the PTCDA molecule and the Al₂O₃ substrate (bond orders, atomic charges)? They are computationally very cheap provided the electronic structure has been optimised. The Authors say: "[...]PTCDA molecule exhibits an increase in charge density" - the above-mentioned analyses would be more quantitative.
- 4) Did the Authors consider the other modes of forming O vacancies?

Figure 3d would be more informative if the threshold density for the isosurface were given.

Some minor language issues (see below) do not lower the general assessment of the reviewed article.

- line 40: "environmental-unfriendly" -> "environmental-unfriendliness",
- line 108: in "To understand the support effect, Al₂O₃, PTCDA, their physical mixture and[...]" please add a comma before "and".
- Line 196: Isn't "butylated hydroxide (BTH)" butylated hydroxytoluene?

Reviewer #2 (Remarks to the Author):

The work entitled "Heterogeneous Organophotocatalytic HBr Oxidation Coupled with Oxygen Reduction for Bromination of Arenes" by Liu and coauthors report that using a supported perylene-3,4,9,10-tetracarboxylic dianhydride (PTCDA) photocatalyst, effective bromination of a series aromatic hydrocarbons can be achieved efficiently. It is found that by only anchoring the organic sensitizer on the alumina support results in a high performance.

This is interesting and extremely crucial for scaling-up applications. The authors have further revealed the insight of the excellent catalytic performance, and point out that the oxygen vacancies in alumina provide Lewis-acid-type anchoring sites for molecular oxygen, enabling a two-electron transfer from the excited PTCDA to oxygen to produce H₂O₂. The rapid reduction of molecular oxygen is a great benefit for the oxidation of HBr into molecular bromine, and eventually the formation of active electrophilic HBrO for bromination. The catalyst and catalytic process are well evidenced by in-situ spectroscopy and theoretical calculations.

The obtained results are solid and will contribute to the development of heterogenization of organophotocatalysts and related applications. I recommend this work for publication in Nature Communications after addressing the following points.

1. During photocatalytic cycles, a new vibrational peak is observed between 1600 and 1700 cm⁻¹ in FT-IR without affecting the catalytic activity. This should be discussed.
2. In the actual reaction process, the reaction is completed under irradiation. However, for the study of the interaction between PTCDA and Al₂O₃, the charge density of PTCDA shows an electron rich state. The change under light conditions should be considered.
3. Other bromine sources, such as NaBr should be analyzed to see if it works.
4. Why Al₂O₃ is used as the substrate? I recommend the authors to consider other wide band-gap semiconductors containing Lewis acid sites as supports.
5. The authors state that Lewis acid sites are the adsorption sites of molecular oxygen. I wonder if there is any correlation between the acidity and the catalytic activity?
6. PTCDA is a well-established red pigment with low price, this makes such process promising for applications. However, I'm still wondering if some derivatives of PTCDA, such as amides, can lead to even better catalytic performances?
7. The detailed operation parameters and procedures of EPR and TPD should be provided.
8. Alkyl halides are synthetically versatile synthons in cross coupling reactions. Can the brominated products produce in this work further couple with other aromatic substrates?
9. Can the brominated products produced in this work undergo C-Br bond cleavage for further hydrogenation or deuterium addition?

Reviewer #3 (Remarks to the Author):

In this work, the heterogeneous organophotocatalyst developed by the authors can dramatically "switch on" the bromination reactivity under mild irradiation condition using HBr as the atom-economic bromination reagent, possessing good extensibility, recyclability and scalability. In which, PTCDA works for photoredox catalyst and oxygen-vacancy-rich Al₂O₃ provides anchoring sites to adsorb O₂, profiting for the two electrons transfer to the adsorbed O₂ for the formation of H₂O₂. The in-situ generated H₂O₂ and Br₂ over redox centers work together to generate HBrO for bromination of arenes. This work is interesting, and with minor revision, acceptance is recommended.

1. The authors proved that strong chemical interaction between PTCDA and Al₂O₃ had been successfully induced by heat treatment via FTIR and XPS, which caused the shift of vibration mode of surface -OH, and also the change in binding energy of anhydride. How these affected the photocatalytic activities?
2. The authors show that the 5%PTCDA/Al₂O₃ has more oxygen vacancy than 5%PTCDA/Al₂O₃(physical mixture). Which, to some extent, suggests that the heat treatment induces the formation of oxygen vacancy. What's the formation mechanism of oxygen vacancy during the heat treatment?
3. From the XRD pattern, the PTCDA displays good crystallinity, why does the HRTEM image of 5%PTCDA/Al₂O₃ not show the lattice fringe of PTCDA? It seems that the PTCDA in the 5%PTCDA/Al₂O₃ is amorphous.
4. The O₂-TPD data is a little bit strange, the authors are suggested to double-check the data.
5. The authors are suggested to double-check the manuscript carefully, some information is missing. For example, the sign of O element mapping in Fig. 1b is missing. The data related to 0.06 mmol Br₂ in Fig. 4d is missing. And the color sign in Fig. S10a is unclear.
6. The languages are suggested to be further checked and polished for better understanding. For example, "Perylene-3,4,9,10-tetracarboxylic dianhydride (PTCDA) is a promising organic photoredox catalyst with proper oxidative potential (1.73 V vs. NHE, pH = 0) and reductive potential (-0.27 V vs. NHE, pH = 0) that may realize simultaneous oxidation of Br⁻ to Br⁺ and reduction of O₂ to reactivexygen species (ROS) for oxidative bromination." Typo "reactivexygen" was discovered.

Responses to Reviewers

Manuscript number: NCOMMS-23-55777

Manuscript Type: Article

Title: Heterogeneous Organophotocatalytic HBr Oxidation Coupled with Oxygen Reduction for Bromination of Arenes

Correspondence Author: Chenliang Su and Bin Liu

Reviewer #1 (Remarks to the Author):

The reviewed article comprises a few novelties:

- Rationalization of the photocatalytic mechanism of the bromination by HBr catalysed by the PTCDA/Al₂O₃ system.
- The heat treatment of the mixture of PTCDA and Al₂O₃ was found necessary for the formation of O vacancies and the formation of the covalent bonding between these subsystems

The Authors competently used many complementary experimental techniques. The experimental part is of high quality. The Authors suggest an unexpected mechanism (dual-electron transfer process to generate H₂O₂ rather than single electron transfer) and their reasoning is well-supported and documented.

In my opinion, the Article deserves publishing in case the below-mentioned questions are addressed.

Response: We are grateful to the reviewer for the positive comments and recommendation for publication of our work in Nature Communications. We also appreciate the reviewer's valuable comments and suggestions, response to which shall greatly improve the quality of our manuscript.

As for the experimental part, a question may emerge about the geometry details of the adsorption mode (see below).

A few objections may be posed for the computational part, namely:

For the adsorption mode of PTCDA on the Al_2O_3 surface:

Comment 1: Al_2O_3 termination - how could the Authors convince the reader that the termination chosen by the Authors is the most represented?

Response 1:

We thank the reviewer for raising this question. Based on high-resolution transmission electron microscopy (HRTEM) and selected area electron diffraction (SAED) characterizations as displayed in Figure R1a, it is deduced that the as-prepared Al_2O_3 is polycrystalline, made of numerous of tiny interconnected nanoparticles. X-ray diffraction (XRD) pattern as shown in Figure R1b indicates that the as-synthesized Al_2O_3 belongs to γ -phase. According to the Gibbs–Curie–Wulff law,¹ for γ - Al_2O_3 , it is estimated that (110) surface is the predominant exposed-surface, which occupies 74% of the total surface area, followed by (100) surface (16%) and (111) surface (10%). As a result, (110) surface was chosen as the active surface to represent the as-prepared Al_2O_3 in our calculation model.

Figure R1. (a) HRTEM image with the corresponding SAED pattern and (b) XRD pattern of the as-prepared Al_2O_3 .

Comments 2: Did the Authors try the other adsorption modes/conformation? If not, what is the reasoning of the chosen one? The HRTEM images (Fig. 1 b) do not show any periodic arrangement.

Response 2:

We thank the reviewer for raising this important question. In fact, other adsorption modes were considered. Experimentally, the adsorption mode of PTCDA was determined by *in-situ* diffuse reflectance infrared Fourier transform spectroscopy (DRIFTS) and X-ray photoelectron spectroscopy (XPS), confirming the interaction between the anhydride groups of PTCDA and the hydroxyl groups of Al_2O_3 . Hence, in theoretical calculation, one or two oxygen atoms of carbonyl group in PTCDA were also considered to interact with the hydroxyl groups on the surface of Al_2O_3 for model 2 and 3 as shown in Figure R2. The calculated adsorption energy is -4.808 eV for model 2 and -3.356 eV for model 3, respectively, more positive than the adsorption energy (-5.744 eV) for model 1 when carbonyl oxygen and C-O-C oxygen co-interact with the hydroxyl groups on the surface of Al_2O_3 . Therefore, model 1 is more thermodynamically favorable.

Model 1

Adsorption energy: -5.744 eV

Model 2

Adsorption energy: -4.808 eV

Model 3

Adsorption energy: -3.356 eV

Figure R2. Different adsorption modes for PTCDA/ Al_2O_3 .

In Figure 1b, there is indeed no periodic arrangement. Al_2O_3 only shows specific crystal planes, while PTCDA is amorphous. This is mainly caused by high-energy electron beam bombardment during HRTEM measurement (Figure R3), consistent with the reported results that PTCDA is crystalline in SAED pattern but amorphous in HRTEM image.²

Figure R3. TEM and HRTEM images of PTCDA.

Comment 3: Did the Authors further study the interaction between the PTCDA molecule and the Al_2O_3 substrate (bond orders, atomic charges)? They are computationally very cheap provided the electronic structure has been optimized. The Authors say: "[...]PTCDA molecule exhibits an increase in charge density" - the above-mentioned analyses would be more quantitative.

Response 3:

We thank the reviewer for the constructive suggestion. According to which, the interaction between PTCDA and the Al_2O_3 substrate was further investigated. Bader charge was calculated (Table R1), from which, it can be clearly observed that there exists significant electron transfer (4.2020 e) from Al_2O_3 to PTCDA (Figure R4). The calculation result is consistent with the experimental XPS measurement result as shown in Figure 3a-c and the differential charge analysis as displayed in Figure 3d.

Figure R4. The differential charge analysis of PTCDA on Al_2O_3 surface. The yellow/cyan region represents increase/decrease in charge density.

Table R1. Atomic layout analysis of PTCDA/ Al_2O_3 .

Atomic species	Number of layout	Effective charge (e)
H1	0.7208	0.2792
H2	0.7456	0.2544
H3	0.8176	0.1824
H4	0.8017	0.1983
H5	0.6926	0.3074
H6	0.7703	0.2297
C1	3.8938	0.1062
C2	4.0799	-0.0799
C3	4.0169	-0.0169
C4	4.0779	-0.0779
C5	4.0478	-0.0478
C6	3.9877	0.0123
C7	4.0105	-0.0105
C8	4.1473	-0.1473
C9	3.9791	0.0209
C10	4.0271	-0.0271

C11	3.9607	0.0393
C12	4.0801	-0.0801
C13	4.011	-0.011
C14	4.0235	-0.0235
C15	3.9972	0.0028
C16	4.0629	-0.0629
C17	4.0412	-0.0412
C18	3.9397	0.0603
C19	4.0705	-0.0705
C20	4.0764	-0.0764
C21	2.9946	1.0054
C22	2.7428	1.2572
C23	2.9709	1.0291
C24	2.747	1.253
O1	7.5439	-1.5439
O2	7.6592	-1.6592
O3	7.5154	-1.5154
O4	7.585	-1.585
O5	7.6241	-1.6241
O6	7.7393	-1.7393

Comment 4: Did the authors consider the other modes of forming O vacancies?

Response 4:

We thank the reviewer for raising this question. NaBH_4 or N_2H_4 reduction method was applied to prepare O vacancy rich Al_2O_3 but failed (Figure R5a). On the other hand, H_2 reduction method could be used to prepare O vacancy rich Al_2O_3 (Figure R5b). Meanwhile, it was found that the photocatalytic activity of 5%PTCDA/ Al_2O_3 (physical mixture), in which Al_2O_3 was prepared by hydrogen reduction treatment, was greatly increased relative to that of PTCDA and Al_2O_3 (Figure R5c), indicating the synergistic effect of PTCDA and oxygen vacancy rich Al_2O_3 in promoting photocatalytic bromination reaction.

Figure R5. (a) EPR signals of Al_2O_3 reduced in NaBH_4 and N_2H_4 . (b) EPR signals and (c) photocatalytic activity of different samples, in which Al_2O_3 was prepared by hydrogen reduction treatment. Reaction conditions: photocatalyst: 10 mg; temperature: 30 °C; time: 5 h; acetonitrile: 1 mL; anisole: 0.2 mmol; HBr: 0.2 mL; blue LED: 100 mW cm^{-2} ; and O_2 : 1 atm.

Comment 5: Figure 3d would be more informative if the threshold density for the isosurface were given.

Response 5:

We thank the reviewer for the nice suggestion. The isovalue is $1.25 \times 10^{-3} \text{ e}\text{\AA}^{-3}$, which has been added to the Figure 3d caption.

Comment 6: Some minor language issues (see below) do not lower the general assessment of the reviewed article.

- line 40: "environmental-unfriendly" -> "environmental-unfriendliness",

- line 108: in "To understand the support effect, Al₂O₃, PTCDA, their physical mixture and[...]" please add a comma before "and".

- Line 196: Isn't "butylated hydroxide (BTH)" butylated hydroxytoluene?

Response 6:

We thank the reviewer for the constructive suggestions. Accordingly, we have made the following changes in the revised manuscript:

1. "environmental-unfriendly" has been changed to "environmental-unfriendliness".
2. A comma has been added in line 108.
3. "butylated hydroxide (BTH)" has been changed to "butylated hydroxytoluene".

Additionally, we have conducted a thorough check of the entire manuscript and made corrections to the grammar mistakes.

Reviewer #2 (Remarks to the Author):

The work entitled “Heterogeneous Organophotocatalytic HBr Oxidation Coupled with Oxygen Reduction for Bromination of Arenes” by Liu and coauthors report that using a supported perylene-3,4,9,10-tetracarboxylic dianhydride (PTCDA) photocatalyst, effective bromination of a series aromatic hydrocarbons can be achieved efficiently. It is found that by only anchoring the organic sensitizer on the alumina support results in a high performance. This is interesting and extremely crucial for scaling-up applications. The authors have further revealed the insight of the excellent catalytic performance, and point out that the oxygen vacancies in alumina provide Lewis-acid-type anchoring sites for molecular oxygen, enabling a two-electron transfer from the excited PTCDA to oxygen to produce H₂O₂. The rapid reduction of molecular oxygen is a great benefit for the oxidation of HBr into molecular bromine, and eventually the formation of active electrophilic HBrO for bromination. The catalyst and catalytic process are well evidenced by in-situ spectroscopy and theoretical calculations. The obtained results are solid and will contribute to the development of heterogenization of organophotocatalysts and related applications. I recommend this work for publication in Nature Communications after addressing the following points.

Response: We are grateful to the reviewer for the positive comments and recommendation for publication of our work in Nature Communications. We also appreciate the reviewer’s valuable comments and suggestions, response to which shall greatly improve the quality of our manuscript. The following list our detailed responses.

Comment 1: During photocatalytic cycles, a new vibrational peak is observed between 1600 and 1700 cm^{-1} in FT-IR without affecting the catalytic activity. This should be discussed.

Response 1:

After five photocatalytic cycles, it was found that the yield of *p*-bromoanisole still exceeded 90%. There showed a new obvious infrared absorption peak at 1663 cm^{-1} , which could be attributed to the stretching vibration of C=O bond, due to partial hydrolysis of anhydride in the presence of HBr.³ This did not change the light absorption property of 5%PTCDA/ Al_2O_3 (Figure R6) as light absorption of PTCDA is mainly resulted from perylene rings. Hence, 5%PTCDA/ Al_2O_3 exhibited stable photocatalytic activity.

Figure R6. UV-vis absorption spectra of 5%PTCDA/ Al_2O_3 before and after five photocatalytic cycles.

Comment 2: In the actual reaction process, the reaction is completed under irradiation. However, for the study of the interaction between PTCDA and Al₂O₃, the charge density of PTCDA shows an electron rich state. The change under light conditions should be considered.

Response 2:

We thank the reviewer for the helpful suggestion. According to the reviewer's suggestion, the distribution of change was further investigated by *in-situ* XPS (Figure R7) under light illumination. It is found that the binding energy of Al 2p XPS peak shifted from 74.4 eV in dark condition to 74.7 eV under light irradiation. The binding energy of C 1s peak attributing to C=O shifted from 286.9 eV in dark condition to 287.6 eV under light irradiation, while the binding energies of C 1s peaks attributing to C=C and C-C did not change significantly. Accordingly, the binding energy of O 1s peak attributing to C=O and C-O shifted from 531.6 eV and 533.5 eV in dark condition to 531.8 eV and 533.7 eV under light irradiation, respectively. These results indicate that photogenerated electrons were transferred from PTCDA to Al₂O₃ under light irradiation.

Figure R7. High-resolution XPS peaks of (a) Al 2p, (b) C 1s and (c) O 1s for 5% PTCDA/Al₂O₃ in dark condition and under light irradiation.

Comment 3: Other bromine sources, such as NaBr should be analyzed to see if it works.

Response 3:

According to the reviewer's suggestion, NaBr was tested as a bromine source for the bromination reaction (Table R2). It was found that no reaction occurred. But the product yield could reach over 95% if HCl solution was added, indicating the importance of H⁺ in the reaction. This is one of the reasons why HBr was chosen as the bromine source in the bromination reaction.

Table R2. Control experiments to study photocatalytic bromination reaction.

Entry	Photocatalysts	Bromine source	Time (h)	Atmosphere	Yield of p -bromoanisole (%)
1	5%PTCDA/Al ₂ O ₃	HBr	5	O ₂	95
2 ^[a]	5%PTCDA/Al ₂ O ₃	NaBr	5	O ₂	0
3 ^[b]	5%PTCDA/Al ₂ O ₃	NaBr + HCl	5	O ₂	96

Reaction conditions: temperature: 30 °C; time: 5 h; acetonitrile: 1 mL; anisole: 0.2 mmol; HBr: 0.2 mL; blue LED: 100 mW cm⁻²; and O₂: 1 atm. [a] 0.4 mmol of NaBr is added instead of HBr in the reaction system. [b] 0.4 mmol of HCl and 0.4 mmol of NaBr are added instead of HBr in the reaction system.

Comment 4: Why Al₂O₃ is used as the substrate? I recommend the authors to consider other wide band-gap semiconductors containing Lewis acid sites as supports.

Response 4:

We thank the reviewer for raising this question. There are two reasons to choose Al₂O₃ as the substrate. Firstly, Al₂O₃ is an insulator, which itself will not contribute to photocatalytic activity. Secondly, Al₂O₃ can be synthesized by a simple hydrothermal method, and its oxygen vacancy content can be modulated by thermal treatment to regulate the adsorption capacity of O₂,⁴ and thus the photocatalytic activity.

According to the reviewer's suggestion, ZrO₂ and Nb₂O₅ (Figure R8) were used to prepare 5%PTCDA/ZrO₂ and 5%PTCDA/Nb₂O₅. It can be observed that the photocatalytic activity of 5%PTCDA/ZrO₂ and 5%PTCDA/Nb₂O₅ can be greatly improved after thermal treatment, but not as much as that of 5%PTCDA/Al₂O₃ (Table R3).

Figure R8. UV-vis absorption spectra of ZrO₂ and Nb₂O₅.

Table R3. Control experiments to study the photocatalytic bromination reaction.

Entry	Photocatalysts	Time (h)	Atmosphere	Yield of p -bromoanisole (%)
1	5%PTCDA/Al ₂ O ₃	5	O ₂	95
2	Al ₂ O ₃	5	O ₂	0
3	PTCDA	5	O ₂	0
4	5%PTCDA/Al ₂ O ₃ (physical mixture)	5	O ₂	0
4	5%PTCDA/Nb ₂ O ₅	5	O ₂	17
5	Nb ₂ O ₅	5	O ₂	0
6	5%PTCDA/Nb ₂ O ₅ (physical mixture)	5	O ₂	0
7	5%PTCDA/ZrO ₂	5	O ₂	9
8	ZrO ₂	5	O ₂	0
9	5%PTCDA/ZrO ₂ (physical mixture)	5	O ₂	0

Reaction conditions: photocatalyst: 10 mg; temperature: 30 °C; time: 5 h; acetonitrile: 1 mL; anisole: 0.2 mmol; HBr: 0.2 mL; blue LED: 100 mW cm⁻²; and O₂: 1 atm.

Comment 5: The authors state that Lewis acid sites are the adsorption sites of molecular oxygen. I wonder if there is any correlation between the acidity and the catalytic activity?

Response 5:

We thank the reviewer for raising this question. To elucidate the relationship between acidity and photocatalytic activity, photocatalysts were prepared using acid and base Al_2O_3 as the substrate. The results showed that the photocatalytic activity of 5%PTCDA/ Al_2O_3 (acid) was significantly higher than that of 5%PTCDA/ Al_2O_3 (base) (Table R4). This indicates that acid Al_2O_3 is beneficial for the photocatalytic bromination reaction.

Table R4. Control experiments to study the photocatalytic bromination reaction.

Entry	Photocatalysts	Time (h)	Atmosphere	Yield of p -bromoanisole (%)
1	5%PTCDA/ Al_2O_3	5	O_2	95
2	5%PTCDA/ Al_2O_3 (acid)	5	O_2	96
3	5%PTCDA/ Al_2O_3 (base)	5	O_2	72

Reaction conditions: Photocatalyst: 10 mg; temperature: 30 °C; time: 5 h; acetonitrile: 1 mL; anisole: 0.2 mmol; HBr: 0.2 mL; blue LED: 100 mW cm^{-2} ; and O_2 : 1 atm.

Comment 6: PTCDA is a well-established red pigment with low price, this makes such process promising for applications. However, I'm still wondering if some derivatives of PTCDA, such as amides, can lead to even better catalytic performances?

Response 6:

We thank the reviewer for raising this interesting question. Perylene diamine (PDI) – a derivative of PTCDA was used to prepare 5%PDI/Al₂O₃. It is found that the photocatalytic activity of 5%PDI/Al₂O₃ is significantly higher than that of single PDI or 5%PDI/Al₂O₃(physical mixture) as shown in Table R5, but not as good as that of 5%PTCDA/Al₂O₃.

Table R5. Control experiments to study the photocatalytic bromination reaction.

Entry	Photocatalysts	Time (h)	Atmosphere	Yield of p -bromoanisole (%)
1	5%PTCDA/Al ₂ O ₃	5	O ₂	95
2	5%PDI/Al ₂ O ₃ (physical mixture)	5	O ₂	0
3	5%PDI/Al ₂ O ₃	5	O ₂	27
4	PDI	5	O ₂	0

Reaction conditions: Photocatalyst: 10 mg; temperature: 30 °C; time: 5 h; acetonitrile: 1 mL; anisole: 0.2 mmol; HBr: 0.2 mL; blue LED: 100 mW cm⁻²; and O₂: 1 atm.

Comment 7: The detailed operation parameters and procedures of EPR and TPD should be provided.

Response 7:

We thank the reviewer for the nice suggestion. The following list the detailed operation parameters and procedures of EPR and TPD.

EPR experiments

To capture $\bullet\text{O}_2^-$, 10 mg of photocatalyst, 1 mL of methanol, and 50 μL of 5,5-dimethyl-1-pyrroline N-oxide (DMPO) solution (methanol : DMPO = 10 : 1) were added into a photocatalytic reactor and then irradiated by visible light under O_2 atmosphere. 20 μL of solution was taken out for EPR measurement after different durations of photocatalytic reaction. To capture $\bullet\text{OH}$, 10 mg of photocatalyst, 1 mL of H_2O , 0.2 mmol of H_2PtCl_6 and 50 μL of 5,5-dimethyl-1-pyrroline N-oxide (DMPO) solution (acetonitrile: DMPO = 10 : 1) were added into a photocatalytic reactor and then irradiated by visible light under Ar atmosphere. 20 μL of solution was taken out for EPR measurement after different durations of photocatalytic reaction. Room-temperature EPR spectra were recorded on a Bruker EMXPLUS10/12 EPR electron paramagnetic resonance spectrometer.

TPD experiments

Oxygen desorption was measured by temperature programmed desorption (TPD) of O_2 in a micro-reactor. Typically, 100 mg of catalyst was added to a micro-reactor, pre-treated in Ar at 200 $^\circ\text{C}$ for 1 h and then cooled to 30 $^\circ\text{C}$. The adsorption of O_2 (30 ml min^{-1}) was implemented for 10 min at 30 $^\circ\text{C}$ and then the catalyst was flushed with Ar (30 ml min^{-1}) for 30 min at 30 $^\circ\text{C}$ to remove the physically adsorbed gas on the surface of the catalyst. Programmed desorption was performed at a heating rate of 10 $^\circ\text{C min}^{-1}$ from 50 to 800 $^\circ\text{C}$ probed by a TCD detector.

Comment 8: Alkyl halides are synthetically versatile synthons in cross coupling reactions. Can the brominated products produce in this work further couple with other aromatic substrates?

Response 8:

We thank the reviewer for raising this question. We have tried the following reaction, but no corresponding product could be obtained.

Comment 9: Can the brominated products produced in this work undergo C-Br bond cleavage for further hydrogenation or deuterium addition?

Response 9:

We thank the reviewer for raising this question. 4-Bromoacetophenone was used as the substrate and triethylamine was used as the hydrogen donor.⁵ It was found that the reaction could not occur. Moreover, 4-bromoacetophenone was used as the substrate and CD₃OD or D₂O was used as the hydrogen donor. It was found that the reaction could not occur either.

Reviewer #3 (Remarks to the Author):

In this work, the heterogeneous organophotocatalyst developed by the authors can dramatically “switch on” the bromination reactivity under mild irradiation condition using HBr as the atom-economic bromination reagent, possessing good extensibility, recyclability and scalability. In which, PTCDA works for photoredox catalyst and oxygen-vacancy-rich Al₂O₃ provides anchoring sites to adsorb O₂, profiting for the two electrons transfer to the adsorbed O₂ for the formation of H₂O₂. The *in-situ* generated H₂O₂ and Br₂ over redox centers work together to generate HBrO for bromination of arenes. This work is interesting, and with minor revision, acceptance is recommended.

Response: We are grateful to the reviewer for the positive comments and recommendation for publication of our work in Nature Communications. We also appreciate the reviewer’s valuable comments and suggestions, response to which shall greatly improve the quality of our manuscript. The following list our detailed responses.

Comment 1: The authors proved that strong chemical interaction between PTCDA and Al₂O₃ had been successfully induced by heat treatment via FTIR and XPS, which caused the shift of vibration mode of surface -OH, and also the change in binding energy of anhydride. How these affected the photocatalytic activities?

Response 1:

We thank the reviewer for raising this question. To study how interaction between PTCDA and Al₂O₃ affected photocatalytic activity, the photocatalysts were prepared at different thermal treatment temperatures, and their photocatalytic activities were measured and compared (Table R6). It can be observed that the photocatalytic activity gradually increased with increasing thermal treatment temperature.

Table R6. Control experiments to study the photocatalytic bromination reaction.

Entry	Photocatalysts	Time (h)	Atmosphere	Yield of p -bromoanisole (%)
1	5%PTCDA/Al ₂ O ₃ -320	5	O ₂	95
2	5%PTCDA/Al ₂ O ₃ -300	5	O ₂	96
3	5%PTCDA/Al ₂ O ₃ -250	5	O ₂	89
4	5%PTCDA/Al ₂ O ₃ -200	5	O ₂	64
5	5%PTCDA/Al ₂ O ₃ -150	5	O ₂	40

Reaction conditions: temperature: 30 °C; time: 5 h; acetonitrile: 1 mL; anisole: 0.2 mmol; HBr: 0.2 mL; blue LED: 100 mW cm⁻²; and O₂: 1 atm.

Comment 2: The authors show that the 5%PTCDA/Al₂O₃ has more oxygen vacancy than 5%PTCDA/Al₂O₃(physical mixture). Which, to some extent, suggests that the heat treatment induces the formation of oxygen vacancy. What's the formation mechanism of oxygen vacancy during the heat treatment?

Response 2:

We thank the reviewer for raising this question. Al₂O₃ was prepared by a hydrothermal method followed by subsequent thermal treatment. It can be observed from the XRD pattern that the crystallinity of the hydrothermal-prepared Al₂O₃ is low. This makes relatively easy breaking of the Al-O bond during thermal treatment, facilitating generation of oxygen vacancies. This result is consistent with the reported literature.⁴

Comment 3: From the XRD pattern, the PTCDA displays good crystallinity, why does the HRTEM image of 5%PTCDA/Al₂O₃ not show the lattice fringe of PTCDA? It seems that the PTCDA in the 5%PTCDA/Al₂O₃ is amorphous.

Response 3:

We thank the reviewer for raising this question. PTCDA displays good crystallinity but does not show periodic lattice arrangement in HRTEM image (Figure R9). This is mainly because of high-energy electron bombardment during HRTEM measurement. This is consistent with the reported results that PTCDA is crystalline in SAED pattern but amorphous in HRTEM image.²

Figure R9. TEM and HRTEM images of PTCDA.

Comment 4: The O₂-TPD data is a little bit strange, the authors are suggested to double-check the data.

Response 4:

We thank the reviewer for the comment. We re-collected the O₂-TPD data (Figure R10). The newly collected data is consistent with our early data as presented in the submitted manuscript.

Figure R10. O₂-TPD curve of 5%PTCDA/Al₂O₃ and the re-collected O₂-TPD curve of 5%PTCDA/Al₂O₃.

Comment 5: The authors are suggested to double-check the manuscript carefully, some information is missing. For example, the sign of O element mapping in Fig. 1b is missing. The data related to 0.06 mmol Br₂ in Fig. 4d is missing. And the color sign in Fig. S10a is unclear.

Response 5:

We thank the reviewer for the constructive comments and suggestions. Accordingly, we have made the following changes:

In Figure 1b, the sign of O element mapping has been added and corrected (as shown in Figure R11).

In Figure 4d, the data related to 0.06 mmol of Br₂ or H₂O₂ has been added and corrected (as shown in Figure R12).

In supplementary Figure 10, the color sign has been corrected (as shown in Figure R13). Additionally, we have carefully checked the entire manuscript and corrected the typos and added the missing information.

Figure R11. Synthesis, characterization and photocatalytic performance. (a) Schematic illustration showing the preparation procedure of PTCDA/Al₂O₃. (b) TEM and HRTEM images, scanning transmission electron microscope (STEM) image and the corresponding elemental mappings of 5%PTCDA/Al₂O₃. (c) XRD patterns of PTCDA/Al₂O₃ with different PTCDA contents. (d) Photocatalytic bromination reaction activities of Al₂O₃, PTCDA, 5%PTCDA/Al₂O₃(physical mixture) and

5%PTCDA/ Al_2O_3 . Reaction conditions: 10 mg of photocatalyst; temperature: 30 °C; reaction time: 5 h; acetonitrile: 1 mL; anisole: 0.2 mmol; HBr: 0.2 mL; O_2 : 1 atm; blue LED: 100 mW cm^{-2} . (e) Evaluation of photocatalytic stability.

Figure R12. The influence of H_2O_2 or Br_2 on photocatalytic bromination reaction. Reaction conditions: temperature: 30 °C; reaction time: 5 h; acetonitrile: 1 mL; anisole: 0.2 mmol; air: 1 atm; blue LED: 100 mW cm^{-2} . In the system of $\text{Br}_2 + \text{H}_2\text{O}$, HBr was not added, but in H_2O_2 system, HBr was added. The amounts of H_2O_2 or Br_2 added are 0.0042, 0.0084, 0.0253, 0.0422 and 0.0843 mmol, respectively.

Figure R13. (a) UV-vis spectra of the solutions for photocatalytic reaction (reaction solution under light with acetonitrile as solvent, 0.00843 mol/L of bromine water and reaction solution under light with acetonitrile as solvent and 10 μL of H_3PO_2 as reducing agent of Br_2 to Br^-). (b) The reaction solution under light with 5%PTCDA/ Al_2O_3 . Reaction conditions: photocatalyst: 10 mg; temperature: 30 °C; time: 5 h; acetonitrile:

1 mL; HBr: 0.2 mL; Ar: 1 atm; H₂PtCl₆: 0.2 mmol; blue LED: 100 mW cm⁻².

Comment 6: The languages are suggested to be further checked and polished for better understanding. For example, “Perylene-3,4,9,10-tetracarboxylic dianhydride (PTCDA) is a promising organic photoredox catalyst with proper oxidative potential (1.73 V vs. NHE, pH = 0) and reductive potential (-0.27 V vs. NHE, pH = 0) that may realize simultaneous oxidation of Br⁻ to Br⁺ and reduction of O₂ to reactivexygen species (ROS) for oxidative bromination.” Typo “reactivexygen” was discovered.

Response 6:

We thank the reviewer for the nice suggestion. The “reactivexygen” has been corrected as “reactive oxygen”. We have carefully checked the entire manuscript and corrected the typos.

References

- 1 Digne, M., Sautet, P., Raybaud, P., Euzen, P. & Toulhoat, H. Use of DFT to achieve a rational understanding of acid-basic properties of γ -alumina surfaces. *J Catal* **226**, 54-68, (2004).
- 2 Chen, G.-F. *et al.* Electrochemical reduction of nitrate to ammonia via direct eight-electron transfer using a copper–molecular solid catalyst. *Nat. Energy* **5**, 605-613, (2020).
- 3 Guo, Y. *et al.* Perylenetetracarboxylic acid nanosheets with internal electric fields and anisotropic charge migration for photocatalytic hydrogen evolution. *Nat. Commun.* **13**, 2067, (2022).
- 4 Li, S.-Q. *et al.* Development of γ -Al₂O₃ with oxygen vacancies induced by amorphous structures for photocatalytic reduction of CO₂. *Chem. Commun.* **58**, 11649-11652, (2022).
- 5 Ghosh, I., Ghosh, T., Bardagi, J. I. & Konig, B. Reduction of aryl halides by consecutive visible light-induced electron transfer processes. *Science* **346**, 725-728, (2014).

REVIEWERS' COMMENTS

Reviewer #1 (Remarks to the Author):

In my comment 1, I asked for the reasoning for the selected termination of the surface. The Authors responded, based on the literature: "As a result, (110) surface was chosen as the active surface[...]", which indeed addresses the issue of the Miller index of the cleavage plane only (which was not questioned) and not the termination I asked about. I assume the Authors also followed the discussion on the termination from the same article cited. If so, I think comment 1 can be regarded as sufficiently discussed.

Comment 3: Even though the Authors added the discussion of the atomic charges, the analysis of the bond orders is still missing. If possible, please add the bond order analysis.

The answers to the other comments seem satisfactory.

Reviewer #2 (Remarks to the Author):

The authors have addressed all my comments properly, and the draft is recommended to publish as is.

Reviewer #3 (Remarks to the Author):

Authors have addressed the issues raised by reviewers. I would like to recommend its publication in the present form.

Responses to Reviewers

Reviewer #1 (Remarks to the Author):

Comment 1: In my comment 1, I asked for the reasoning for the selected termination of the surface. The Authors responded, based on the literature: "As a result, (110) surface was chosen as the active surface[...]", which indeed addresses the issue of the Miller index of the cleavage plane only (which was not questioned) and not the termination I asked about. I assume the Authors also followed the discussion on the termination from the same article cited. If so, I think comment 1 can be regarded as sufficiently discussed.

Response: In our article, we indeed followed the discussion on the termination of the surface from the cited reference.¹ We thank the reviewer again for raising the constructive questions.

Comment 3: Even though the Authors added the discussion of the atomic charges, the analysis of the bond orders is still missing. If possible, please add the bond order analysis.

Response: We thank the reviewer for the nice suggestion. The bond order can be defined by molecular orbital theory as: half of the total number of electrons that occupy the bonding molecular orbitals minus the number of electron pairs that occupy the antibonding molecular orbitals. Bond order analysis helps in quantifying the strength and nature of chemical bonds between atoms, providing insights into the bonding interactions involved in the separation process. The popular bond analyses are Wiberg,² Mayer,³ and Nalewajski-Mrozek (NM).⁴ At present, these computational methods are mainly applicable to organic molecules and organic ligand models, and not applicable to organic molecules and inorganic materials with periodic structures. For our system, to further analyze the interaction between PTCDA and Al₂O₃, the bonding/antibonding population is analyzed using the projected crystal orbital Hamilton population (pCOHP), which, can also indicate the strength of bonding.⁵ Integral COHP values are -2.13324 for Model 1, -1.85124 for Model 2, and -1.44866 for Model 3 as displayed in Figure R1, demonstrating stronger interaction between PTCDA and Al₂O₃ in Model 1. COHP analysis has been added into the revised Supplementary Information (as shown in Supplementary Figure 6).

Figure R1. Negative-projected COHP ($-p\text{COHP}$) for the O and H atoms with the

shortest bond length in three different adsorption configurations for PTCDA/Al₂O₃(110).

The answers to the other comments seem satisfactory.

Response: We are grateful to the time and efforts made by the reviewer on assessing our work.

Reviewer #2 (Remarks to the Author):

The authors have addressed all my comments properly, and the draft is recommended to publish as is.

Response: We are grateful to the reviewer for the positive comments and recommendation for publication of our work in Nature Communications.

Reviewer #3 (Remarks to the Author):

Authors have addressed the issues raised by reviewers. I would like to recommend its publication in the present form.

Response: We are grateful to the reviewer for the positive comments and recommendation for publication of our work in Nature Communications.

References

- 1 Digne, M., Sautet, P., Raybaud, P., Euzen, P. & Toulhoat, H. Use of DFT to achieve a rational understanding of acid-basic properties of γ -alumina surfaces. *J. Catal.* **226**, 54-68, (2004).
- 2 Wiberg, K. B. Application of pople-santry-segal complete neglect of differential overlap (CNDO) method to some hydrocarbons and their cations. *J. Am. Chem. Soc.* **90**, 1083-1096, (1968).
- 3 Mayer, I. Bond order and valence indices: A personal account. *J. Comput. Chem.* **28**, 204-221, (2007).
- 4 Nalewajski, R. F., Mrozek, J. & Michalak, A. Two-electron valence indices from the Kohn-Sham orbitals. *In. J. Quantum Chem.* **61**, 589-601, (1997).
- 5 Maintz, S., Deringer, V. L., Tchougréeff, A. L. & Dronskowski, R. LOBSTER: a tool to extract chemical bonding from plane-wave based DFT. *J. Comput. Chem.* **37**, 1030-1035, (2016).